# COVID-19 self-testing: Countries accelerating policies ahead of WHO guidelines during pandemics, a global consultation

**Melody Sakala**[1,2]*, **Cheryl Johnson**[3]*, **James Chirombo**[1,2], **Jilian A. Sacks**[3], **Rachel Baggaley**[3], **Titus Divala**[1]

**1** Malawi Liverpool Wellcome Programme, Blantyre, Malawi, **2** Kamuzu University of Health Sciences, Blantyre, Malawi, **3** World Health Organisation, Geneva, Switzerland

* msakala@mlw.mw (MS); johnsonc@who.int (CJ)

**Data Availability Statement:** Data is available on DOI:10.6084/m9.figshare.23993484.

## Abstract

The widespread use of antigen-detection rapid diagnostic tests (Ag-RDTs) has revolutionized SARS-CoV-2 (COVID-19) testing, particularly through the option of self-testing. The full extent of Ag-RDT utilization for self-testing, however, remains largely unexplored. To inform the development of WHO guidance on COVID-19 self-testing, we conducted a global consultation to gather the views and experiences of policy makers, researchers, and implementers worldwide. The consultation was conducted by disseminating a WHO questionnaire through professional networks via email and social media, encouraging onward sharing. We used a cross-sectional design with both closed and open-ended questions related to policy and program information concerning the regulation, availability, target population, indications, implementation, benefits, and challenges of COVID-19 self-testing (C19ST). We defined self-testing as tests performed and interpreted by an untrained individual, often at home. Descriptive summaries, cross-tabulations, and proportions were used to calculate outcomes at the global level and by WHO region and World Bank income classifications. All information was collated and reported according to WHO guideline development standards and practice for global consultations. Between 01 and 11 February 2022, 844 individuals from 139 countries responded to the survey, with 45% reporting affiliation with governments and 47% operating at the national level. 504 respondents from 101 countries reported policies supporting C19ST for a range of use cases, including symptomatic and asymptomatic populations. More respondents from low-and-middle-income countries (LMICs) than high-income countries (HICs) reported a lack of an C19ST policy (61 vs 11 countries) and low population-level reach of C19ST. Respondents with C19ST experience perceived that the tests were mostly acceptable to target populations, provided significant benefits, and highlighted several key challenges to be addressed for increased success. Reported costs varied widely, ranging from specific programmes enabling free access to certain users and others with high costs via the private sector. Based on this consultation, systems for the regulatory review, policy development and implementation of C19ST appeared to be much more common in HIC when compared to LIC in early 2022, though most respondents indicated self-testing was available to some extent (101 out of 139 countries) in their country. Addressing such global inequities is critical for ensuring access to innovative and impactful

**Funding:** This work was supported by UNITAD and the funders had no role in study design, data collection, analysis, decision to publish or preparation of the manuscript.

interventions in the context of a public health emergency of international concern. The challenges and opportunities highlighted by key stakeholders could be valuable to consider as future testing strategies are being set for outbreak-prone diseases.

## Introduction

The emergence of SARS-CoV-2, the virus responsible for COVID-19, was declared a pandemic by the World Health Organization (WHO) in January 2020 [1]. Since then, COVID-19 has caused social, human, and economic crises globally, leading to an accelerated drive for the development of interventions, including diagnostic testing [2]. In particular, the pandemic put unprecedented strain on global health systems, leading to an urgent need for rapid and accessible testing modalities particularly in low- and middle-income countries (LMICs). This resulted in the widespread adoption of antigen-based rapid diagnostic tests (Ag-RDTs), which can affordably provide same-day results, early diagnosis, and prompt referrals to onward services and care, including quarantine or treatment, to protect the most vulnerable [3]. C19ST was thought to reduce absenteeism at work, made people feel safer and increased uptake and identification of new cases [3].

Despite ongoing transmission of SARS-CoV-2 globally, confirmed cases represent only a fraction of the number of infections, with access to testing being a major factor influencing case confirmation rates [4]. While laboratory-based polymerase chain reaction (PCR) tests were initially considered the gold standard for COVID-19 diagnostic testing, the development of (Ag-RDTs) provided an accurate, cost-effective and easily accessible alternative [5].

The advent of portable, easy-to-use, and stable Ag-RDTs has rapidly expanded access to testing, opened the possibility of self-testing, and expanded the scope and potential of self-care. Self-testing has been an important intervention recommended by WHO across many different disease areas, including HIV, viral hepatitis, and STIs, particularly in low- and middle-income countries [6,7]. Self-testing has consistently been shown to increase access to and uptake on testing overall and can greater uptake of both prevention and treatment services [8,9]. While the use of self-testing is increasingly available and well-accepted [10,11], it is important to understand the experiences and views in the context of COVID-19. We conducted a global consultation targeting policy makers, researchers, and implementers to generate an understanding of policies and experiences with SARS-CoV-2 Ag RDTs for self-testing. Our findings informed the development of WHO guidelines and serve as a future reference point as global policies and practices continue to evolve.

## Methodology

### Study design

We conducted a global consultation as part of the development of WHO guidelines on COVID-19 self-testing using a self-administered electronic questionnaire from 1st-11th February 2022.

### Setting and participants

Our target participants were policy makers, implementers, and academics from all WHO member states. The questionnaire was disseminated globally by the WHO to gather responses from a diverse range of participants, including through country and regional office focal

points, through social media [e.g. Facebook, twitter, LinkedIn] and professional networks and associations. To ensure a wide distribution, we invited participants to engage in the consultation through professional networks via email and social media and encouraged onward sharing.

### Data collection tool

We developed the consultation questionnaire in SurveyMonkey, which included questions relevant to participants' understanding or perceptions of current policy, practice, and progress of implementing COVID-19 AgRDT in their settings. The questionnaire focused on AgRDT use in general, and we restricted our analysis to questions that focused on self-testing. We designed questions to focus on the following areas: current availability of C19ST regulatory processes, policy, and use, target population for C19ST, national coverage and willingness of the target population to undertake C19ST, requirements and mechanisms for reporting C19ST results according to policy or practice, mechanisms and costs for accessing C19ST kits according to policy or practice, and benefits and challenges of implementing C19ST.

### Key variables

We defined self-testing as tests performed and interpreted by an untrained individual, typically at home. Respondents provided their characteristics, including country of operation, affiliation, and level of involvement in the COVID-19 response. We asked stakeholders both closed and open-ended questions related to policy and program information concerning the regulation, availability, target population, indications, implementation, benefits, and challenges of COVID-19 self-testing. Participants answered questions they were able to and not all respondents answered all questions.

### Statistical methods

We summarized the data using both tabular and graphical descriptive statistics. For tables, we reported the number (n) of responses for each question without missing values, the size of the analysis population (N) and calculated the proportions (n/N). We prepared cross-tabulations and produced grouped bar charts of key outcomes to show proportions stratified by variables such as WHO region and country income level classification. We reported the data at a global level and by WHO region and income classifications. We also performed some analyses at the individual respondent level, and others by at the country level regardless of the number of individuals. We conducted all data cleaning and analyses using the R language for statistical computing.

### Ethics statement

The findings reported are from a global consultation which was initiated as part of the development of WHO guidelines on COVID-19 self-testing. As such, it was carried out in accordance with the WHO Handbook for Guideline Development https://apps.who.int/iris/handle/10665/145714 and aligns with standards used in previous published WHO consultations [12]. The development of all WHO guidelines is overseen by the WHO Guideline Review Committee. While the project was not set up as a formal research study, a strict protocol was followed according to which the purpose of the consultation was explained at the beginning of each interview, including that results would be shared as part of a global decision-making process and may be published. Written informed consent was obtained before starting the consultation. All participants were informed that they could stop the interview at any point.

This consultation was conducted with a strong commitment to ethical considerations. The cross-sectional survey design used, disseminated globally via online platforms, ensured voluntary participation. The purpose of the consultation was clear to all participants. Some identifiable data was collected; however, participants' anonymity was strictly maintained during data management analysis and reporting; and no individual identities will be disclosed or compromised.

## Results

### Respondents' characteristics

From 1 to 11 February 2022, 844 individuals participated in the consultation across 139 countries. Table 1 describes the characteristics of the participants, including their representation across all WHO regions and income classifications.

Out of all participants, 45% identified as government officials, and 47% reported involvement in implementing SARS-CoV-2 activities at the national level.

### Current availability of C19ST regulatory processes, policy and use

Table 2 outlines the availability of C19ST regulatory processes, policy, and use, as reported by 294 participants from 92 countries across all regions at the time of the survey (early Feb 2022).

Most participants indicated that their countries had a regulatory process in place for reviewing and approving C19ST kits. Specifically, 32% (278) of participants from 81 countries reported that their countries had a C19ST policy in place or was actively developing one.

**Table 1. Characteristics of consultation participants.**

| Number of participants(n) | 831 | 100% |
|---|---|---|
| **WHO Region of Participant's Country** | | |
| African | 229 | 27.2% |
| Americas | 80 | 9.5% |
| Eastern Mediterranean | 64 | 7.6% |
| European | 176 | 20.9% |
| South-East Asian | 120 | 14.3% |
| Western Pacific | 173 | 20.5% |
| **World Bank Income Classification of Country** | | |
| Low income | 74 | 9% |
| Lower middle income | 424 | 51% |
| Upper middle income | 134 | 16% |
| High income | 199 | 24% |
| **Level Of Involvement in the COVID-19 Response** | | |
| National level | 390 | 47% |
| Regional/provincial/District level | 223 | 27% |
| Community level | 135 | 16% |
| Other | 83 | 10% |
| **Affiliation** | | |
| Academic | 85 | 10% |
| Government | 378 | 45% |
| NGO | 167 | 20% |
| Private sector | 92 | 11% |
| Other | 109 | 13% |

**Table 2. Reported availability of C19ST regulatory processes, policy, and use.**

| | n | Does your country have a policy on C19ST? | | | | | Are SARS-CoV-2 Ag-RDTs used for self-testing in your country? | |
| --- | --- | --- | --- | --- | --- | --- | --- | --- |
| | | Policy in place, Supportive | Policy in place, not supportive | No policy, but Piloting | Developing policy | No policy, no pilot | Yes (n/N) | percent |
| All respondents | 481 | 29% | 5% | 11% | 17% | 37% | 259/481 | 54% |
| **WHO Region of Respondents' Country** | | | | | | | | |
| African | 137 | 15% | 2% | 7% | 21% | 55% | 30/137 | 22% |
| Americas | 35 | 34% | 3% | 6% | 6% | 51% | 25/36 | 69% |
| Eastern Mediterranean | 39 | 21% | 5% | 18% | 13% | 44% | 24/38 | 63% |
| European | 78 | 49% | 8% | 14% | 10% | 19% | 67/77 | 87% |
| South-East Asian | 80 | 31% | 8% | 14% | 8% | 40% | 39/82 | 48% |
| Western Pacific | 112 | 33% | 4% | 13% | 30% | 20% | 74/114 | 65% |
| **World Bank Income Classification of Respondents' Country** | | | | | | | | |
| Low income | 92 | 12% | 4% | 12% | 12% | 60% | 23/96 | 24% |
| Lower middle income | 241 | 24% | 4% | 13% | 25% | 34% | 121/247 | 49% |
| Upper middle income | 57 | 28% | 4% | 9% | 19% | 40% | 35/58 | 60% |
| High income | 79 | 59% | 5% | 9% | 3% | 24% | 72/83 | 87% |

Notably, a lack of policy was reported more frequently by participants from low and middle-income countries (LMIC) (25.2%) than by those from high-income countries (HIC) (10.1%). Participants from HIC were more likely to report that C19ST was used in their country compared to those from LMIC (88% vs 19%).

## Target population for C19ST

251 participants from 15 countries reported targeting four specific populations for COVID-19 self-testing. Among participants who indicated that there was a range of different populations noted as being targeted for self-testing, reported that this included testing symptomatic individuals, as well as asymptomatic testing in workplaces, educational institutions and to support attendance at larger gatherings.

## National coverage and willingness of target population to undertake C19ST

Nationwide implementation of C19ST was reported by 148 participants from 79 countries, of which 34% were high-income and 65% were LMIC (13.5% Low income, 27.8% = Lower Middle, 21.5% = Upper Middle).

In Table 3, the majority of participant's noted that less than half the national population is estimated to have used C19ST, with the highest perception of use reported by respondents from the European region and in High-Income Countries. Participants from HICs reported more widespread use of C19ST than those from low- and middle-income countries (LMICs). For both LMICs and HICs, 33.3% reported C19ST national coverage between 20 to 50%. In LMICs, 23.5% reported a C19ST reach of less than 1%, while in HICs only 4.8% reported a less than 1% coverage of C19ST. Most stakeholders across all regions perceived that the target populations were either very willing or willing to use self-tests. Willingness to self-test was consistent between high- and low-income country categories as shown in Table 3 above.

**Table 3. Participant's perception of national coverage and willingness of target population to use self-tests.**

| | | What proportion of your national population do you estimate has used C19ST? | | | | | Based on your experience, how willing are the target populations to use C19ST? | | | | |
|---|---|---|---|---|---|---|---|---|---|---|---|
| | | n | <1% | 1–20% | 20–50% | >50% | n | Very willing | Willing | Somewhat willing | Not at all willing |
| All respondents (n = 134) | | 134 | 19% | 33% | 29% | 19% | 152 | 41% | 30% | 25% | 3% |
| **WHO Region of Participant's Country** | | | | | | | | | | | |
| African | | 15 | 27% | 20% | 33% | 20% | 18 | 44% | 50% | 6% | 0% |
| Americas | | 9 | 11% | 56% | 11% | 22% | 9 | 44% | 22% | 33% | 0% |
| Eastern Mediterranean | | 10 | 60% | 10% | 20% | 10% | 14 | 50% | 7% | 43% | 0% |
| European | | 30 | 0% | 13% | 40% | 47% | 34 | 38% | 38% | 15% | 9% |
| South-East Asian | | 29 | 28% | 48% | 17% | 7% | 31 | 35% | 23% | 42% | 0% |
| Western Pacific | | 41 | 15% | 41% | 34% | 10% | 46 | 43% | 30% | 22% | 4% |
| **World Bank Income Classification of Participant's Country** | | | | | | | | | | | |
| Low income | | 16 | 50% | 31% | 13% | 6% | 17 | 29% | 41% | 29% | 0% |
| Lower middle income | | 67 | 22% | 43% | 25% | 9% | 74 | 41% | 24% | 31% | 4% |
| Upper middle income | | 17 | 6% | 24% | 65% | 6% | 19 | 37% | 37% | 26% | 0% |
| High income | | 34 | 3% | 18% | 26% | 53% | 38 | 47% | 34% | 13% | 5% |

## Requirements and mechanisms for reporting C19ST results according to policy and practice

Figs 1 and 2 present results on requirements and mechanisms for reporting C19ST results according to policy or practice.

Reporting of positive C19ST results was more common, but it was not required in some regions. In both high- and low-income countries, mechanisms for reporting results included in person, by phone, mobile application, and using a website.

## Distribution methods mechanisms and costs for accessing C19ST kits according to policy and practice

Fig 3 outlines the distribution methods of self-testing kits, as reported by 222 respondents.

The most common distribution method for C19ST kits was through pharmacies (47.3%).

Table 4 shows that direct consumer costs for purchasing C19ST kits varied substantially across respondents and by country, but 58 respondents from 31 countries indicated that C19ST was free for some end-users. Free access to C19ST was reported more frequently in high-income countries than in low-income countries (37.1% vs 18.1%).

## Perceived benefits, challenges, and areas of improvement

Participants reported several challenges related to implementing C19ST, including insufficient infrastructure for distribution and storage of C19ST, poor understanding of how to use and interpret the tests, lack of training for personnel, and lack of awareness among the public about the availability and benefits of C19ST. Participants were also asked about potential solutions to the challenges identified. Many stakeholders recommended increased training and education on C19ST, including proper usage and interpretation of results, as well as addressing issues related to distribution and storage of C19ST. Other suggestions included increasing public awareness about the availability and benefits of C19ST, making C19ST more affordable or providing free access, and increasing international cooperation to address the global shortage of C19ST. Other benefits of implementing C19ST that were mentioned included increased

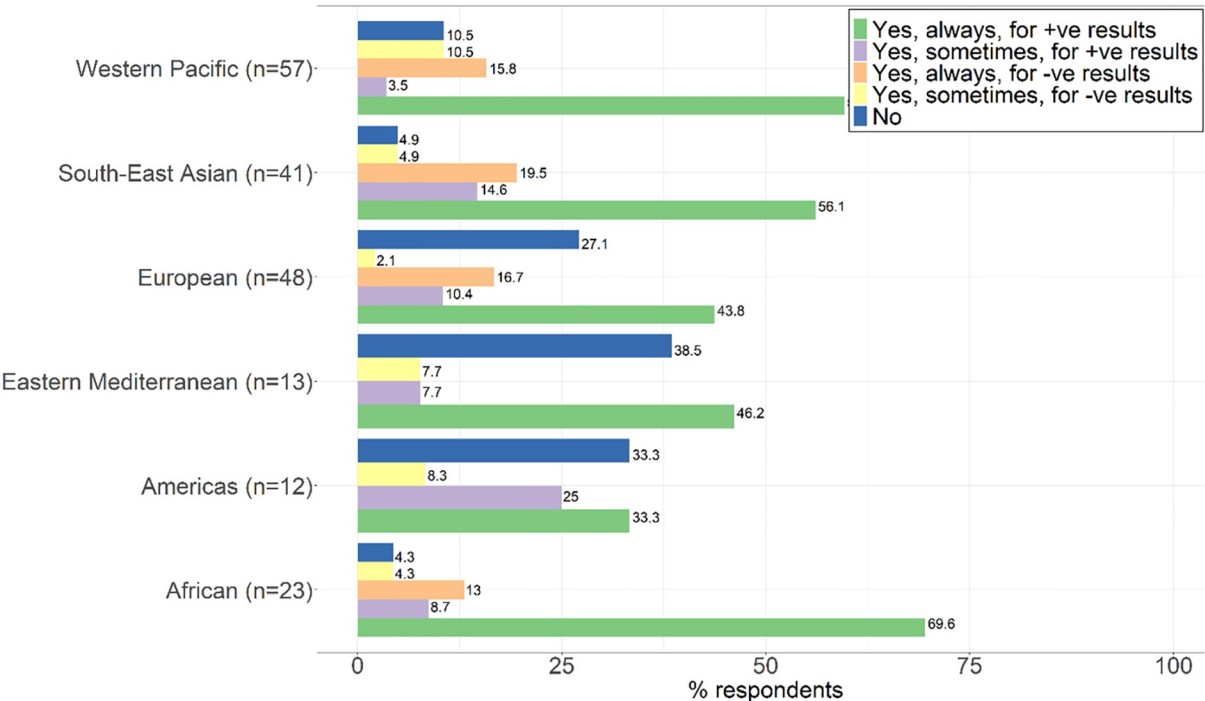

**Fig 1. Requirements for reporting C19ST results.**

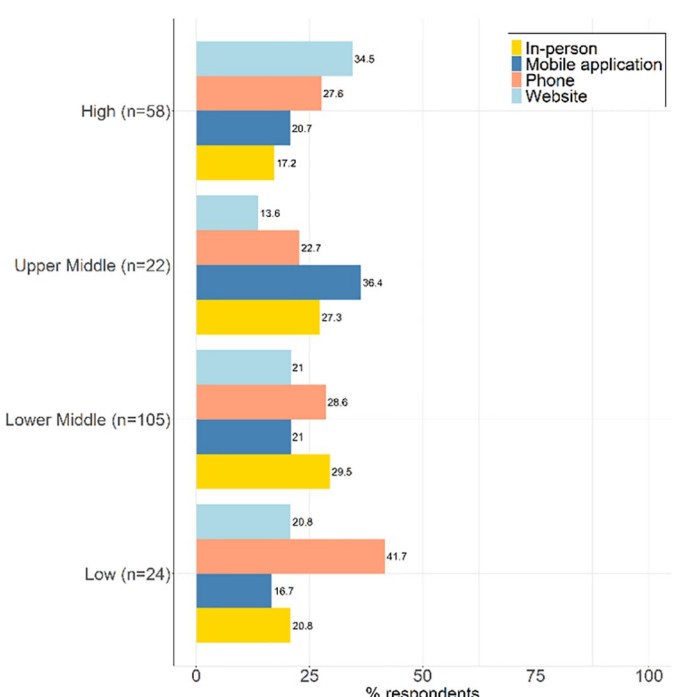

**Fig 2. Mechanisms for reporting results.**

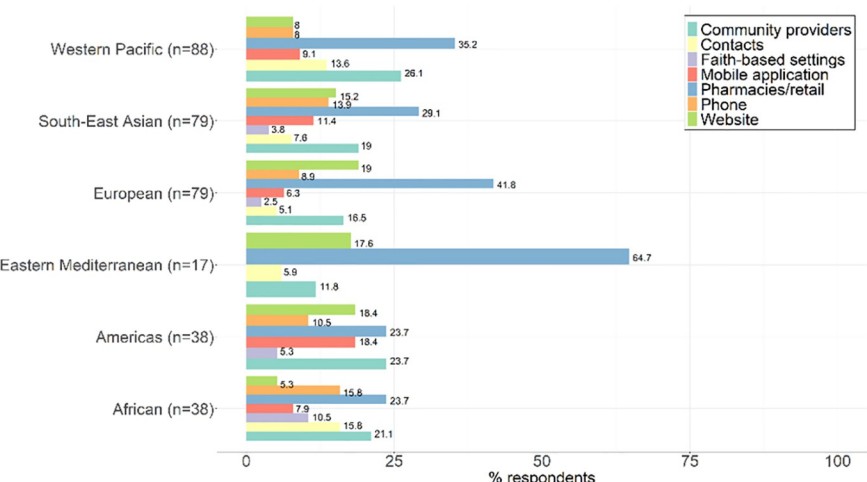

**Fig 3. Distribution methods for C19ST.**

confidence in identifying and controlling outbreaks, increased accuracy in identifying cases, and potential reduction in the burden on healthcare systems.

## Discussion

This global consultation assessed the extent of policy and implementation of COVID-19 self-testing according to perceptions of targeted policy makers, implementers, and researchers contributing to various levels of COVID-19 response as of early 2022. Good participation globally; supportive policy and more wide use reported in HIC compared to LIC or LMIC–also differences across regions. On the other hand, there was high willingness to use self-tests irrespective of income classification. Wide range of distribution approaches reported with pharmacy/retail generally the most frequent; reported costs ranged widely with free tests more available in HICs than in LMICs. The reported cost of accessing C19ST kits was variable, ranging from free to very costly (e.g. over USD 20). Reporting requirements and mechanisms also varied–generally increased expectation to report positive results but a high proportion of respondents did note no known reporting requirements. Supportive policy across a range of use cases was overwhelmingly reported by most countries globally.

At the time of the consultation there were no WHO guidelines on C19ST, our study played a role in highlighting the global inequalities in access to COVID-19 interventions, with wider

**Table 4. Cost for accessing C19 self-testing kits.**

| Cost | < 1$ | 1–2$ | 2–5$ | <5–10$ | 20–20$ | 20+$ |
|------|------|------|------|--------|--------|------|
| Countries | Malawi | South Korea | Portugal | Canada | Australia | Russia |
| | Indonesia | Malaysia | Vietnam | Peru | Ukraine | Philippines |
| | | Pakistan | India | Mauritius | Mexico | Papua New Guinea |
| | | | Netherlands | Spain | | United states of America |
| | | | Belgium | Cyprus | | |
| | | | Thailand | Finland | | |
| | | | Singapore | Qatar | | |
| | | | | Bahrain | | |

coverage and access to C19ST reported in high-income countries compared to low- and middle-income countries. An alternative explanation to the global differences in C19ST utilisation between the global South and North could be lack of awareness and policy. Europe, which has the highest proportion of population with access, had the highest proportion of respondents reporting availability of policy. This highlights the role of a WHO guideline in increasing global awareness and utilisation of interventions. The recently released global guidance on C19ST is expected to support development of national policies.

While self-testing can be beneficial as it enables access without requiring action from overstretched professional health workers; and can be more convenient without requiring individuals to go to a health care facility. The widely reported supportive policy and use of C19ST across a range of settings is likely to reflect the acceptable diagnostic performance [13,14] and potential for scalability of this rapid testing technology, which are critical to the COVID-19 pandemic response. COVID-19 Ag RDTs can be produced much faster, cheaper and in large quantities for large scalability. Antigen tests offer the possibility of rapid, inexpensive detection of SARS-COV-2 [15]. Community engagement with relevant stakeholders is crucial to successful COVID-19 implementation and scale up [6,16,17]. Such community-led approaches have been found to be highly effective when adapted for HIV self-testing [18], and should be considered and adapted as part of future pandemic preparedness plans [19].

The consultation showed a wide range of target populations in both asymptomatic and symptomatic individuals and in either non healthcare, education institution, and mass gatherings settings using C19 ST. The wide range of target populations within and across regions demonstrates wide applicability and the ease of use in place of professional testing. The versatility of C19ST is critical during the COVID-19 pandemic when health systems are stretched for human resource. This is especially important for low-income countries where the diagnostic testing gap is widest in part secondary to financial and human resource challenges. Self-testing using Ag-RDTs can be utilised for triage or identifying individuals with the highest viral load [20].

Willingness to self-test was consistent across region and income setting. The willingness of target populations to self-test is encouraging but not surprising because of the individual and community benefits of C19ST. Unlike professional testing, which may require a visit to a health facility, C19ST like most self-care interventions, allows users to know their status within 30 minutes and without leaving their home. C19ST therefore limits contact with other people and speeds up commencement of post-test actions, including those related to protecting the health of the user, their family and their community. Moderate or severely symptomatic individuals may quickly see the need to access COVID-19 hospital care. Individuals who test negative, seamlessly resume usual activities, or decide to seek alternative clinical care if sick. Other studies support high perceived willingness to use C19 self-testing [7,21,22] however recent consultations and epidemiological developments may be contributing to a decline in willingness over-time.

The consultation found wider coverage and access use of C19 self-testing from high Income countries compared to low- and middle-income countries. The striking differences in coverage and access to C19ST between high- and low-income nations reflects substantial global inequalities with respect to access to COVID-19 interventions. The strong correlation between GDP and testing capacity was initially driven by the centralised nature and high capital requirements of PCR testing. While C19ST offers the opportunity for wider access based on the affordability of AgRDTs, the survey results suggest that there are still some underlying barriers to full utilisation that need to be addressed in low-income countries. To address this gap, greater efforts are needed moving forward to ensure LMICs have access to critical innovations and interventions during public health emergencies and pandemics.

The perceived challenges highlighted by participants including users' inability to perform or interpret C19ST, false positive results, false negative results, people being forced to self-test, failure to report results, and failure to present for confirmatory testing must be taken into consideration when developing national C19ST strategies or policies. To date, studies resonate similar challenges on self-testing observed [8,16] from other disease programs, particularly HIV. To address the main barrier to perform or accurately translate results, involving the community in co-creation and dissemination of education materials would be beneficial. In addition, some literature recommends inclusion of health promotion, layman's language and public sensitization to increase implementation success of C19ST use [23,24] while users in other studies recommended locally translated and easily accessible materials through online or in person [8]. As with other self-testing approaches, programmatic support including engagement with health workers, community groups and those most affected are needed to achieve the greatest impact. Future strategies using C19ST should consider how to adapt materials from other self-testing programs and services. This may improve both acceptability and feasibility of implementation, as well as maximize limited resources.

Although the consultation provides valuable insights into COVID-19 policies and practices, it is essential to consider its limitations. The consultation findings do not necessarily reflect the official positions of countries and ministries of health but rather presents the perspectives of a diverse group of respondents involved in COVID-19 work across countries. Additionally, the findings represent a snapshot of policy and practice as of February 2022 and may not reflect earlier or later circumstances as the pandemic and response continue to evolve. Nevertheless, these findings importantly highlight the views and experiences with C19ST from many countries experiencing outbreaks and reporting high case rates. Thus, the findings may be applicable for future tracking of changes in policies and practices and be helpful for planning in the event of outbreaks or seasonal spread.

The use of Ag-RDTs for self-testing is a promising solution, but our finding of limited access in low-and-middle-income countries compared to high-income countries highlights the harsh reality of global inequalities and the need to address them sustainably. Experiences shared by respondents highlighted benefits and challenges and opportunities that can inform development of effective policies to support self-testing at national and global levels. Moving forward as part of pandemic preparedness, global stakeholders need to prioritize investment to enable LMICs to benefit from critical innovations and interventions.

## Acknowledgments

We would like to thank all participants in the consultation. We acknowledge WHO guideline development group and colleagues for their untiring feedback throughout the whole process.

## Author Contributions

**Conceptualization:** Cheryl Johnson, Jilian A. Sacks, Rachel Baggaley, Titus Divala.

**Data curation:** James Chirombo.

**Formal analysis:** Melody Sakala, James Chirombo, Titus Divala.

**Methodology:** Melody Sakala, Cheryl Johnson, James Chirombo, Jilian A. Sacks, Titus Divala.

**Software:** James Chirombo.

**Supervision:** Cheryl Johnson, Jilian A. Sacks, Rachel Baggaley, Titus Divala.

**Writing – original draft:** Melody Sakala, Titus Divala.

**Writing – review & editing:** Melody Sakala, Cheryl Johnson, James Chirombo, Jilian A. Sacks, Titus Divala.

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
