## [Decision Letter · Decision Letter 0]

23 Oct 2023

PGPH-D-23-01568

COVID-19 self-testing: Countries accelerating policies ahead of WHO guidelines during pandemics: A Global Survey.

Dear Dr. Sakala,

Thank you for submitting your manuscript to PLOS Global Public Health. After careful consideration, we feel that it has merit but does not fully meet PLOS Global Public Health’s publication criteria as it currently stands. Therefore, we invite you to submit a revised version of the manuscript that addresses the points raised during the review process.

EDITOR: Please insert comments here and delete this placeholder text when finished. Be sure to:

Indicate which changes you require for acceptance versus which changes you recommendAddress any conflicts between the reviews so that it's clear which advice the authors should followProvide specific feedback from your evaluation of the manuscript

Please ensure that your decision is justified on PLOS Global Public Health’s publication criteria and not, for example, on novelty or perceived impact.

We look forward to receiving your revised manuscript.

Kind regards,

Saskia Popescu, PhD

Academic Editor

Journal Requirements:

Additional Editor Comments (if provided):

Major revision is needed to address the gaps in data collection and survey methods - ethical approval, etc. This is an important study and would like to see it published, but please make sure to address critical information gaps within it.

Reviewers' comments:

Reviewer's Responses to Questions

**Comments to the Author**

1. Does this manuscript meet PLOS Global Public Health’s publication criteria? Is the manuscript technically sound, and do the data support the conclusions? The manuscript must describe methodologically and ethically rigorous research with conclusions that are appropriately drawn based on the data presented.

Reviewer #1: No

2. Has the statistical analysis been performed appropriately and rigorously?

Reviewer #1: Yes

3. Have the authors made all data underlying the findings in their manuscript fully available (please refer to the Data Availability Statement at the start of the manuscript PDF file)?

Reviewer #1: Yes

4. Is the manuscript presented in an intelligible fashion and written in standard English?

Reviewer #1: Yes

5. Review Comments to the Author

Reviewer #1: How Verbal informed consent was obtained via this online survey. Please explain how and in what context was this consent obtained. This manuscript is very descriptive in nature and designed for a policy decision for a WHO working committee and not intended to be devised as a research study. There were no ethical committee approval obtained for this survey to be claimed as a research study.

6. PLOS authors have the option to publish the peer review history of their article (what does this mean?). If published, this will include your full peer review and any attached files.

**Do you want your identity to be public for this peer review?** For information about this choice, including consent withdrawal, please see our Privacy Policy.

Reviewer #1: No

---

## [Editor Report · Decision Letter 1]

2 Feb 2024

COVID-19 self-testing: Countries accelerating policies ahead of WHO guidelines during pandemics: A Global Survey.

PGPH-D-23-01568R1

Dear Miss Sakala,

We are pleased to inform you that your manuscript 'COVID-19 self-testing: Countries accelerating policies ahead of WHO guidelines during pandemics: A Global Survey.' has been provisionally accepted for publication in PLOS Global Public Health.

Best regards,

Saskia Popescu, PhD

Academic Editor